# Multidrug Resistant Coagulase-Positive *Staphylococcus aureus* and Their Enterotoxins Detection in Traditional Cheeses Marketed in Banat Region, Romania

**DOI:** 10.3390/antibiotics10121458

**Published:** 2021-11-26

**Authors:** Adriana Morar, Alexandra Ban-Cucerzan, Viorel Herman, Emil Tîrziu, Khalid Ibrahim Sallam, Samir Mohammed Abd-Elghany, Kálmán Imre

**Affiliations:** 1Department of Animal Production and Veterinary Public Health, Faculty of Veterinary Medicine, Banat’s University of Agricultural Sciences and Veterinary Medicine “King Michael I of Romania” from Timişoara, 300645 Timisoara, Romania; adrianamo2001@yahoo.com (A.M.); alexandra_cucerzan@yahoo.com (A.B.-C.); emiltarziu@yahoo.com (E.T.); 2Department of Infectious Diseases and Preventive Medicine, Faculty of Veterinary Medicine, Banat’s University of Agricultural Sciences and Veterinary Medicine “King Michael I of Romania” from Timişoara, 300645 Timisoara, Romania; viorel.herman@fmvt.ro; 3Department of Food Hygiene and Control, Faculty of Veterinary Medicine, Mansoura University, Mansoura 35511, Egypt; khalidsallam@mans.edu.eg (K.I.S.); drsamir@mans.edu.eg (S.M.A.-E.)

**Keywords:** coagulase-positive staphylococci, multidrug resistant *S. aureus*, staphylococcal enterotoxins, traditional cheeses

## Abstract

The main objectives of the present study were to determine the occurrence of coagulase positive staphylococci (CPS) and to assess the presence and antimicrobial susceptibility profile of *Staphylococcus aureus* isolates in different raw milk origin (cow and sheep) traditional cheeses marketed in Banat region, Romania. Additionally, the presence of *mecA* gene in *S. aureus* isolates and the staphylococcal enterotoxins (SEs) in cheese samples were evaluated. A total of 81.6% (138/169) of the screened samples were positive for CPS. Furthermore, 35.5% (49/138) of the investigated CPS positive cheese samples were contaminated with *S. aureus*, with an isolation frequency of 46.6% (14/30) in caș, 33.3% (32/96) in telemea, 25% (2/8) in burduf, and 25% (1/4) in urdă assortments, respectively. From the total number of *S. aureus* isolates, 6.1% (3/49) harbored the *mecA* gene. Detectable levels of SEs were identified in 4.3% (4/94) of cheese samples with a CPS contamination level higher than 10^5^ log CFU g^−1^. The expressed antimicrobial susceptibility profile of the tested cheese-origin *S. aureus* isolates, with the automated Vitek 2 equipment, showed resistance towards amikacin (90.1%, 10 out from 11 tested), enrofloxacin (86.2%, 25/29), ceftiofur (72.7%, 8/11), neomycin (63.6%, 7/11), benzylpenicillin (53.1%, 26/49), kanamycin (41.4%, 12/29), rifampicin (39.5%, 15/38), tetracycline (38.8%, 19/49), tilmicosin (36.4%, 4/11), clindamycin (30.6%, 15/49), ciprofloxacin (30%, 6/20), erythromycin (22.4%, 11/49), tylosin (18.2%, 2/11), oxacillin (16.3%, 8/49), linezolid (15%, 3/20), teicoplanin (15%, 3/20), fusidic acid (13.1%), imipenem (10.5%, 4/38), vancomycin (7.9%, 3/38), ampicillin (5.5%, 1/18), mupirocin (5.5%, 1/18), fosfomycin (5%, 1/20), and gentamicin (4.1%, 2/49). Twenty-four (49%) *S. aureus* isolates exhibited multidrug resistance. The investigation highlighted a common occurrence of multidrug-resistant *S. aureus* strains in the monitored cheese assortments, which can constitute a potential risk for consumers’ health.

## 1. Introduction

*Staphylococcus aureus* subsp. *aureus,* the main representative species of the coagulase-positive staphylococci, is a Gram positive, nonmotile, facultative anaerobic, ubiquitous bacteria throughout nature. This microorganism, having natural ecological niches in the nasal cavity and skin of warm-blooded animals, is considered a leading cause of several health-care-associated infections (e.g., skin and soft tissue abscesses, endocarditis, osteomyelitis, bacteraemia). In addition, its toxic shock syndrome and food poisoning effect in humans, resulting from the ingestion of the produced thermostable and pepsin-resistant enterotoxins in a certain food matrix, has also been well documented [1]. In milk producing animals, *S. aureus* often colonizes the ductus papillaris of the mammary glands, which can result in the frequent appearance of clinical or subclinical mastitis phenomenon, generating reasons for antimicrobial use in dairy industry [2].

The production of traditional raw milk cheeses in the backyard and/or small-scale integrated livestock at farm level, and their direct sale to the final consumer within agri-food markets has a long and very popular cultural heritage in Romania, passed from generation to generation, offering a decent income for farmers. The microbiological safety of these products is closely linked to the successful implementation of food safety management systems within the production–storage–dispatch–marketing chain by farmhouse cheesemakers, as opposed to industrially produced cheese assortments whose safety is guaranteed by the manufacturer through authorized veterinary personnel. Nevertheless, several recently conducted studies in the surrounding Balkan Peninsula region, in countries with traditional cheese manufacturing history [3,4,5,6], demonstrated that raw milk cheese can harbor enterotoxin producing *S. aureus* strains. In addition, in the last decade, the involvement of the raw milk cheese origin staphylococcal toxins in food-borne outbreaks in several European Union (EU) member states (e.g., France, Italy, Hungary) has been frequently highlighted in the annually published European Food Safety Authority (EFSA) and European Centre for Disease Prevention and Control (ECDC) scientific reports [7].

Nowadays, the foodborne antimicrobial resistance (FAMR) phenomenon is an increasing biological hazard, as a consequence of irrational decades-long usage of antimicrobials. Based on the One Health concept, to prevent the propagation of drug-resistant bacteria and to achieve better public health outcomes, an integrated AMR surveillance system along the food chain is recommended [7]. In this regard, in Romania, limited information is available concerning the study of the occurrence and antimicrobial susceptibility profile of the methicillin-resistant *S. aureus* (MRSA) strains in different food products [8,9], demonstrating the potential role of the screened food matrices in the transmission, colonization, and infection of the pathogen lineages among humans. 

Considering the lack of any available scientific peer reviewed publications, the present survey aimed to determine the occurrence of coagulase positive staphylococci (CPS), and to assess the presence and antimicrobial susceptibility profile of *S. aureus* isolates in different raw milk origin traditional cheeses, marketed in Banat region, Romania. Additionally, the presence of *mecA* gene in *S. aureus* isolates and the staphylococcal enterotoxins in cheese samples were evaluated. 

## 2. Results

### 2.1. Frequency of Isolation and Contamination Level of CPS, and Incidence of S. aureus

Of the 169 traditional raw milk origin cheese samples examined, 138 (81.6%; 95% CI 75.1–86.7) were contaminated with CPS, with a distribution of 101 (83.5% from the total of 121; 95% CI 75.8–89.0) and 37 (77.1% from the total of 48; 95% CI 63.5–86.7) positive samples in the periods of February 2012–October 2013, and January 2020–February 2021, respectively. Concerning the examined cheese categories, the highest frequency of isolation was recorded in telemea samples (88.1%; 96/109, 95% CI 80.6–92.9), followed by caș (78.9%; 30/38, 95% CI 63.6–88.9), burduf (72.7%; 8/11, 95% CI 43.4–90.3), and urdă (36.4%; 4/11, 95% CI 15.1–64.6) assortments. These differences are not statistically significant (*p* > 0.05). The detailed distribution of CPS in cheese samples according to the milk origin (sheep or cow), as well as the registered minimum, maximum, and mean values in each assortment are presented in Table 1.

Of the 138 CPS positive cheese samples, 49 (35.5%, 95% CI 28.0–43.8) were contaminated with *S. aureus*. Out of them, 38 (37.6% from the total of 101; 95% CI 28.8–47.4) samples were obtained in the study period February 2012–October 2013, and another 11 (29.7% from total of 37; 95% CI 17.5–45.8) samples in the period January 2020–February 2021, with an isolation frequency of 37.6% and 29.7%, respectively. Their overall distribution according to the tested assortments were 46.6%, 33.3%, 25.0%, and 25.0% in caș, telemea, burduf, and urdă samples, respectively (Table 1). These differences are not statistically significant (*p* > 0.05).

### 2.2. Prevalence of Genetic Markers among the Isolated S. aureus Strains

Both *16S rDNA* and *nuc* genes were detected in all (49) *S. aureus* isolates tested, confirming the results of their biochemical testing with the Vitek2 system. From the total number of isolates, three (6.1%; 95% CI 2.1–16.5) harbored the *mecA* gene, being recovered from two (8.3%) telemea and one (16.6%) caș assortments in the period of January 2020–February 2021 (Table 1). These isolates were accounted as methicillin-resistant, because the detection of this gene represents the standard procedure in the determining resistance towards methicillin. It is important to mention that in the case of MRSA strains, from the total genome, ~25% can consist on the presence of mobile genetic elements (e.g., bacteriophages, plasmids, transposoms, pathogenecity islands, or chromosomal cassettes) constituting the so-called accessory genome, responsible for genetic material transfer between strains (reviewed by [1]).

### 2.3. Detection of SEs in Cheese Samples

Detectable levels of SEs were identified in four (4.3%, 95% CI 1.7–10.4) out of 94 tested cheese samples with a CPS contamination level higher than 10^5^ log CFU g^−1^ (Table 1). All of these positive findings were obtained in the study period of February 2012–October 2013.

### 2.4. Antimicrobial Resistance Profile of S. aureus Isolated Strains

Antimicrobial resistance profile of the tested *S. aureus* strains (*n* = 49), according to their isolation sources, are summarized in Table 2. In addition, the antimicrobial susceptibility test results according to the used specific bacteria cards, are available in the Appendix A.

Twenty isolates tested with the AST-P592 card expressed resistance in descending order towards RIF (50%), PCG (40%), CIP (30%), CLI (25%), TEC (25%), OXA (20%), FA (20%), IPM (15%), ERY (15%), LZD (15%), TET (15%), VAN (10%), and FOF (5%), but all the isolates were susceptible to GEN, MXF, SXT, and TGC (Table 2, Appendix A). Furthermore, the exhibited antimicrobial susceptibility profile of another 18 isolates screened with the AST-GP69 card showed that the resistance to ENR (77.8%) was the most common, followed by that to TET (44.4%), PCG (38.9%), RIF (27.8%), KAN (11.1%), OXA (5.6%), VAN (5.6%), AMP (5.6%), IPM (5.6%), FA (5.6%), and MUP (5.6%), but none of the isolates were resistant to SAM, GEN, MBX, ERY, CLI, NIT, CHL, and SXT (Table 2, Appendix A). Concerning the monitoring of drug resistance of 11 *S. aureus* isolates using the AST-GP79 card, resistance was observed towards PCG (100%), ENR (100%), KAN (90.9%), AMK (90.9%), CLI (90.9%), CTF (72.7%), ERY (72.7%), TET (72.7%), NEO (63.6%), TMS (36.4%), OXA (27.3%), TYL (18.2%), and GEN (18.2%), whereas total susceptibility was recorded for AMP, CET, CEF, FLO, and SXT (Table 2, Appendix A). 

Twenty-four (49%, 95% CI 35.6–62.5) out from 49 tested *S. aureus* isolates exhibited MDR (telemea origin *S. aureus* strains to 3–8 classes; caș—to 3–7 classes) resulting in the expression of a total of 20 resistance profiles (Table 3), while four (8.2%) strains were susceptible to all tested drugs. No notable associations (*p* < 0.05) were registered between the expressions of the antimicrobial resistance patterns of the tested *S. aureus* isolates and their isolation sources.

## 3. Discussion

To the authors’ knowledge, this is the first extended study in Romania investigating the presence and public health significance of the CPS in traditional cheese assortments destinated for human consumption. As has been highlighted in several previously conducted investigations, CPS could be isolated from the raw milk origin cheese assortments by a wide range of frequencies and counts. Thus, the recorded prevalence for CPS contamination in the present study (81.6%) is lower than that reported in neighboring Serbia (87.4%; [4]), but higher than that obtained in Poland (69.2%; [11]), Sweden (69%; [12]), Italy (23.7%; [13]), or in another recent conducted survey in Serbia (20.5%; [6]). Variations of the recorded results can be related upon by the microbiological quality of the raw milk and level of hygienic measures in the farmhouse establishments during cheese production. The recorded relatively high frequency of isolation of CPS in the present study, reflects the low control measures applied during the cheese-making process by producers, including handling of the curd without gloves, face mask or head cover [4,11,12]. In addition, within the main obtaining technological steps (e.g., milk coagulation, curd processing, pressing or early ripening) the unrestrictive direct exposure of the assortments to the environment (e.g., air, dust, food contact surfaces) can greatly contribute to the increasing of the amount of mesophilic halotolerant staphylococci in the final product [4]. 

Ninety-four (68.1% 95% CI 59.9–75.3) samples harbored CPS levels above the regulatory limit of >10^5^ established by the European Regulation (EC) No. 2073/2005 modified by the Regulation (EC) No. 1441/2007 [14]. Accordingly, in case of such contamination level SEs must be screened. As has been previously demonstrated, the low generation time of staphylococci (0.8 h at 25 °C) resulting in their rapid growth in the milk, favored by their physical entrapment by the curd can be considered explanations for the recorded high contamination level [10]. These findings demonstrate the unsuccessful implementation of adequate hygiene measures by the majority of manufacturers, resulting in potentially hazardous products to consumers. 

Similar to our results, other *S. aureus* surveys conducted in different cheese types yielded positive findings in Croatia (54%; [15]), Scotland (33%; [16]), Poland (69.2%; [11]), and Brazil (70%; [17]). As has been previously concluded, these bacteria may be introduced in the raw milk and subsequently in dairy products in several ways such us (*i*) directly from the udder level with clinical or subclinical staphylococcal mastitis, (*ii*) accidental fecal contamination, (*iii*) milking equipment or (*iv*) skin of the personnel involved in cheese production [18,19]. The resultant overall frequency of isolation rate of *S. aureus*, able to produce heat stable toxins in food, strengthens the need of an integrated surveillance system at the level of production units to avoid the transmission of this pathogen within traditional cheeses.

The *mecA* gene is the genotypic determinant of MRSA strains, encoding the PBP2a synthesis which phenotypically reflect in MIC values higher than 4 µg/mL for oxacillin [20]. In agreement with our results, positive findings, with relatively low detection rate of the *mec*A gene in cheese origin *S. aureus*, were reported in studies conducted in Iran (16.2%, [21]), Italy (1.3%, [22]; 3.8%, [23]) or Switzerland (0.2%, [24]). Contrary, a number of investigations reported that none of the tested *S. aureus* isolates were *mecA* positive [4,25,26]. The finding of the present survey pointed out that the consumption of raw milk origin traditional cheeses in the screened area can constitute a potential risk for the acquiring of MRSA strains, with subsequent possible challenges for public health specialists.

Even if the ELFA-Vidas^®^ *Staph enterotoxin II* method is recognized as a sensitive assay [27], contrary to our results, other studies reported negative findings in the detection of SEs in cheese samples contaminated with a high level of CPS [11,28]. The variations in toxin detection may be related to the existence/absence of appropriate environmental conditions (e.g., temperature) [11]. Nevertheless, the obtained results provide an objective assessment of the real toxicity risk for human health.

This is the first report about the antimicrobial susceptibility testing of the cheese origin *S. aureus* isolates. Overall, the expressed resistance pattern of the tested strains revealed increased resistance towards some drugs and decreased or total susceptibility to others. Concerning the seven antimicrobials, included in all of the used three specific cards, with which all of the *S. aureus* isolates were tested with different MIC range values, the highest resistances were observed towards PCG (53.1%) and TET (38.8%). Similar to our results, considerable AMR trends were recorded for penicillin in Serbia (66.6%, [4]) and Poland (50.8%, [11]); and for TET in Iran (56.1%, [21]) and China (98.1%, [29]). In addition, even if the recorded resistance level for CLI (30.6%), ERY (22.4%), OXA (16.3%) and GEN (4.1%) can be accounted as moderate or relatively low, the study findings indicated the possible over-usage of these drugs in the Romanian livestock veterinary medicine and an urgent adaptation of an efficient antimicrobial stewardship programs. Contrary to our results, in other studies a total susceptibility has been noted for CLI [4], OXA [29], and ERY [30].

Unfortunately, 90.1% (10 out from 11 tested isolates), 86.2% (25/29), 72.7% (8/11), 63.6% (7/11), 41.4% (12/29), 39.5% (15/38), and 36.4% (4/11) of a certain number of tested isolates showed resistance to AMK, ENR, CTF, NEO, KAN, RIF, and TMS consecutively (Table 2). These results are comparable to those reported in other investigations [4,11,21,29,30]. The recorded increased resistance patterns can be considered another proof of the undesired outcomes of the intensive and irrational use of these drugs to control and treat infections in milk producing animals. Furthermore, these findings are in correlation with antibiotics that are regularly used for treatment of livestock infections in Romania (Imre, unpublished results).

The recorded high number of MDR *S. aureus* isolates (49%) is another notable finding of the present survey. Usually, the use of a drug to which a bacterium has expressed an intermediate level of resistance in the management of infections caused by that bacterium is associated with an uncertain therapeutic effect. Also, the intermediate category signals the need for a higher dose than the standard one, favoring the development of antibiotic resistance [1]. Based on these considerations and to avoid undesired outcomes of the treatment of *S. aureus* infections by physicians and veterinary practitioners, in the present study the obtained intermediate resistance levels in the case of some drugs (RIF, CTF, NEO, CIP, MUP, TEC) were considered as resistant (Table 2). In this regard, it is important to mention that the consideration of intermediate MIC results as resistant towards a certain drug resulted in the reporting of more resistance within the tested *S. aureus* strains. Our results, in general, confirm the steady upward trend of AMR phenomenon reported in other earlier studies for food and/or animal origin strains. Thus, in the neighboring Serbia, 25.0% of the tested isolates expressed MDR and the most frequently encountered resistance profiles included PCG, CIP, or GEN [4]. In another investigation, a total of 12.8% of the screened milk and dairy products origin *S. aureus* isolates were MDR [21]. In addition, in context of the One Health approach, the recorded worrying 100% MDR pattern in case of *S. aureus* strains isolated from shelter dogs with skin lesions from Timișoara Municipility, included in the monitored Banat region in the present study, indirectly suggests the increased AMR trend in the case of this pathogen in Romania [31]. In contrast, in a study conducted in Poland most of the isolates (47.5%) were found to be resistant to only one antimicrobial [11].

Three (7.9%) out from 38 tested *S. aureus* strains in the present study were resistant to VAN. This worrying result derives from the fact that this glycopeptide antibiotic, during the last decades, has been considered to be a mainstay first line drug in the treatment of invasive MRSA infections. The emergence of the complete vancomycin-resistant *S. aureus* (VRSA) strains, conferred by the *vanA* gene, has been identified, for the first time, in 2002 in Michigan, USA [32]. To date, a total of 52 VRSA strains carrying the *vanA* gene have been reported worldwide (reviewed by [33]). It is important to mention that the acquiring of VAN resistance in the case of *S. aureus* can be closely related to the presence of the *vanA* gene clusters and several mobile genetic elements (e.g., self-transferable or mobilizable plasmids, conjugative transposons) in other Gram-positive bacteria (e.g., *Enterococcus faecalis*, *E. faecium*, *Clostidium difficile*) [34]. Until now, VAN resistance has not been reported for food origin *S. aureus* strains in Romania, but its occurrence has been recently confirmed for strains isolated from shelter dogs with skin lesions [31]. Further molecular studies focusing on the evidence of the *van*A gene in a large number of food or animal origin *S. aureus* isolates are still required to improve our understanding about the demonstrated spreading of VAN resistance by this pathogen in Romania.

## 4. Materials and Methods

### 4.1. Sample Collection

A total of 169 Romanian traditional raw milk cheese specimens, providing from small scale integrated dairy production units and exposed directly for sale, were collected between February 2012 and October 2013, and January 2020 and February 2021, from nine agri-food markets located in urban settlements of the historical Banat region (NW—46° 6’39.38”N, 20°16’37.12”E; NE—45°56’55.70”N, 22°17’40.37”E; SE—44°46’35.53”N, 22°31’27.89”E; SW—45°21’18.06”N, 21° 0’2.92”E), South-Western part of Romania. The total number of samples to be collected (*n* = 169) was determined using the proposed formula by Cochran (1963), with 87.4% expected prevalence, 5% absolute precision (confidence level), and 95% confidence interval (CI) [35]. Also, sample collection followed the requirement of the European Regulation (EC) No. 2073/2005, meaning that for each sample, a pool of the randomly selected five products (approx. ~300 g from each), from the same category and retailer were collected. The selected agri-food markets were located in the two county (Timiș and Caraș-Severin) residences of the assessed region, namely Timișoara (*n* = 6) and Reșița (*n* = 3) municipalities, being visited daily by hundreds of consumers. The samples were aseptically collected by the research team members, in the first month of each trimester of the calendar year, accompanying the veterinary authorities during the officially organized control visits. The raw milk cow and/or sheep origin cheese samples (see their distribution in Table 1) included the following four assortments: telemea (*n* = 109)—white brined soft cheese; caș (*n* = 38)—semisoft, unsalted fresh cheese; burduf (*n* = 11)—slightly soft kneaded cheese; and urdă (*n* = 11)—fresh whey cheese. The collected samples were individually placed into a sterile disposable plastic bag and labeled with the assortment name, sampling date and location. In the same day, the samples were transported under refrigeration conditions (≤4 °C) to the laboratory of Food Hygiene and Microbiological Risk Assessment of the Faculty of Veterinary Medicine, Timișoara for microbiological analysis.

### 4.2. Isolation and Enumeration of Coagulase-Positive Staphylococci (CPS) and Identification of S. aureus

The isolation and enumeration of CPS was performed following the International Organization and Standardization (ISO) 6888-1:2021 standard with slight modifications [36]. In brief, 10 g of each sample was homogenized in a Stomacher (bioMérieux, Marcy l’Etoile, France) (90 s) with 90 mL of preheated (45 °C) peptone-buffered solution (PBS; pH = 7.5 ± 0.1). Next, samples were serially diluted in sterile 0.5% peptone water up to 10^−4^ (1:10.000) dilution. Aliquots of 0.1 mL from each dilution were further cultured on the selective and differential Baird–Parker agar supplemented with egg yolk tellurite emulsion (Oxoid, Basingstoke, Hampshire, UK) medium at 37 °C for 36–48 h. The samples which produced typical colonies (i.e., convex black circular colonies surrounded by a transparent and/or opaque halo), of which number were counted, were considered to contain CPS. The number of typical colonies was assessed using the manual counting of colony-forming (CFU) units’ technique, recommended for estimating the number of viable bacteria cells. Five colonies from each positive plate were randomly selected, transferred and cultured on Brain Heart Infusion Agar (Biokar Diagnostics, Allone, France) at 37°C for 24 h. After growth, the presumptive *Staphylococcus* spp. isolates were examined using Gram staining, coagulase, and catalase test. Subsequently, isolates were identified at species level by testing their complete biochemical properties with the Vitek2 automated compact system (bioMérieux, Marcy l’Etoile, France), and using the Vitek 2^®^ ID-GP (Gram-positive) specific identification cards, following the manufacturer guidelines.

### 4.3. Molecular Analyses

One *S. aureus* isolate per positive sample were directly subjected to molecular processing. Bacterial genomic DNA was isolated using a PureLink^TM^ Genomic DNA Mini Kit (Invitrogen™, Carlsbad, CA, USA) following the manufacturer’s instructions. Firstly, molecular characterization of the isolates was accomplished through a genus-specific conventional uniplex polymerase chain reaction (PCR), targeting the *16S rDNA* gene (~886 bp) and using the 16S-1 (5’-GTGCCAGCAGCCGCGGTAA-3’) and 16S-2 (5’-AGACCCGGGAACGTATTCAC-3’) primer set. Next, in order to evidence the *nuc* gene (~255 bp), specific to *S. aureus* encoding a thermostable nuclease (TNase), another PCR protocol was carried out implicating the specific nuc-1 (5’-TCAGCAAATGCATCACAAACAG-3’) and nuc-2 (5’-CGTAAATGCACTTGCTTCAGG-3’) primers. Subsequently, the presence of *mecA* gene (533 bp), which confers resistance to methicillin encoding the penicillin-binding protein 2a (PBP2a) of *S. aureus*, were screened using the specific *mecA*-1 (5’-GGGATCATAGCGTCATTATTC-3’) and *mec*A-2 (5’-AACGATTGTGACACGATAGCC-3’) primer set. PCR reaction mix and cycling conditions was established as indicated by Poulsen et al. [37]. Within each PCR assays the *S. aureus* ATCC 25923^TM^ *mecA* positive strain was used as positive control, and PCR grade water (no template DNA) as negative control. The resultant PCR products were visualized on Midori Green (Nippon Genetics^®^; Europe, Gmbh, Düren, Germany) stained 1.8% agarose gel.

### 4.4. Enterotoxin Detection

Cheese samples (*n* = 94) with CPS contamination level higher than 10^5^ log CFU g^−1^, being considered necessary for assumptive enterotoxin production [38], were screened for the presence of classical staphylococcal enterotoxins (SEs) using the Vidas^®^ Staphylococcal enterotoxin II (SET2, bioMérieux, Marcy l’Etoile, France) assay, performed with the Mini VIDAS (bioMérieux, Marcy l’Etoile, France) automated system according to the manufacturer instructions. The methodology, based on Enzyme Linked Fluorescent Assay (ELFA) technique using anti-SEs antibodies, allows the simultaneous detection of seven SEs serotypes (A, B, C_1_, C_2_, C_3_, D and E), with a limit of detection of 0.25 ng/g of food matrix. The system automatically quantifies the results as positive or negative.

### 4.5. Antimicrobial Susceptibility Testing

Antimicrobial susceptibility testing of the cheese origin *S. aureus* isolates (*n* = 49) was performed with the fully automated Vitek2 equipment (bioMérieux, Marcy l’Etoile, France). In this regard, three different Gram-positive specific bacteria cards were used. Each of them was selected in a certain interval of the study period, depending on their commercial availability and aiming to cover both livestock and human origin antimicrobials. Thus, the AST-P592, AST-GP69, and AST-GP79 Gram positive specific bacteria cards were used in the testing of the antimicrobial susceptibility profile of 20, 18, and 11 isolates, respectively (Appendix A). These cards included a total of 33 antimicrobials (amikacin [AMK], ampicillin [AMP], ampicillin/sulbactam [SAM], benzylpenicillin [PCG], cefalotin [CET], cefquinome [CEF], ceftiofur [CTF], chloramphenicol [CHL], ciprofloxacin [CIP], clindamycin [CLI], enrofloxacin [ENR], erythromycin [ERY], florfenicol [FLO], fosfomycin [FOF], fusidic acid [FA], gentamicin [GEN], imipenem [IPM], kanamycin [KAN], linezolid [LZD], marbofloxacin [MBX], moxifloxacin [MXF], mupirocin [MUP], neomycin [NEO], nitrofurantoin [NIT], oxacillin [OXA], rifampicin [RIF], teicoplanin [TEC], tetracycline [TET], tigecycline [TGC], tilmicosin [TMS], trimethoprim-sulfamethoxazole [SXT], tylosin [TYL], vancomycin [VAN]), with a correspondent minimum inhibitory concentration [MIC] range, from 15 classes (aminoglycosides, amphenicols, fluoroquinolones, glycopeptides, lincomycins, macrolides, nitrofuran derivates, oxazolidinones, phosphonic acid derivatives, pseudomonic acid derivates, quinolones, rifamycins, β-lactams, steroids, sulfonamides and tetracyclines), as indicated in the Appendix A, respectively. The isolates were categorized by the Vitek2 equipment as susceptible, intermediate or resistant to the tested antimicrobials. The isolates tested intermediate to a certain drug were considered resistant, and the MDR phenomenon was quantified based on the acquired resistance to at least one antimicrobial in three or more antimicrobial classes [39]. Resistance breakpoints were established according to the Clinical Laboratory Standards Institute (CLSI) guideline [20]. Within determinations, the internal quality control was performed with the *S. aureus* ATCC 29213^TM^ strain.

### 4.6. Statistical Analysis

The obtained data were statistically analyzed using the SPSS software version 20.0 (IBM, Armonk, NY, USA). Differences in the frequency of isolation of *S. aureus* strains in relation to their sample origin were estimated using the Pearson’s chi-square test and were considered significant at *p* ≤ 0.05.

## 5. Conclusions

The results of the present study have increased our knowledge about the frequency of isolation of CPS, *S. aureus* and SEs in different traditional Romanian cheeses in the screened region, reflecting hygienic deficiencies during their manufacturing. The occurrence of multidrug-, methicillin-, and vancomycin-resistant *S. aureus* strains, with variable detection rate in the monitored cheese assortments, indicates a substantial public health and medical concern, and the possible over-use of some drugs in the livestock breeding industry. Likewise, the findings could reflect the increased monitoring of the biological safety of these products by veterinary authorities, and can provide useful insight for veterinarians and physicians in the management of *S. aureus* infections. The present study offers baseline information on the knowledge of the phenotypic antimicrobial resistance of food origin *S. aureus* strains in the screened region. However, further studies based on molecular tools, investigating the presence of other antibiotic resistance genes and mobile genetic elements responsible for genetic material transfer between *S. aureus* strains, are still necessary to obtain a better knowledge of the complex puzzle of the AMR phenomenon of this important food-borne pathogen in Romania.

## Figures and Tables

**Table 1 antibiotics-10-01458-t001:** Presence of coagulase positive staphylococci (CPS), *S. aureus* and staphylococcal enterotoxins in the investigated traditional raw milk origin cheeses in Banat region, Romania.

Cheese Assortments and Their Origin (No. of Examined)		No. of Samples Containing CPS (%)	CPS Levels of Contamination (log CFU/g^−1^)							No. of CPS Positive Samples Containing *S. aureus* (%)	No. of *S. aureus* with *mecA* Gene (%)	No. of Cheese Samples with Enterotoxins/Tested (%)
			Below the Regulatory [10] Limit ≤ 10^5^			Above the Regulatory Limit > 10^5^			Mean ± SD			
			Min.	Max.	No. (%)	Min.	Max.	No. (%)				
**telemea**	sheep milk (*n* = 77)	71 (92.2)	3.26	4.91	11 (15.5)	5.51	7.12	60 (84.5)	5.67 ± 0.85	24 (33.8)	2 (8.3)	1/60 (1.7)
	cow milk (*n* = 32)	25 (78.1)	2.37	4.81	18 (72.0)	5.00	7.14	7 (28.0)	4.19 ± 1.40	8 (32.0)	-	1/7 (14.3)
**caș**	sheep milk (*n* = 20)	17 (85.0)	4.15	4.62	4 (23.5)	5.38	7.06	13 (76.5)	5.66 ± 0.92	6 (35.3)	1 (16.6)	2/13 (15.4)
	cow milk (*n* = 18)	13 (72.2)	2.61	4.75	6 (46.2)	5.26	6.36	7 (53.8)	4.80 ± 1.19	8 (61.5)	-	0/7
**burduf**	sheep milk (*n* = 11)	8 (72.7)	4.61	4.80	3 (37.5)	5.41	7.26	5 (62.5)	5.83 ± 1.07	2 (25.0)	-	0/5
**urdă**	sheep milk (*n* = 11)	4 (36.4)	3.62	4.48	2 (50.0)	6.86	7.32	2 (50.0)	5.57 ± 1.79	1 (25.0)	-	0/2
**Overall (*n* = 169)**		138 (81.6)	N.A.	N.A.	44 (31.9)	N.A.	N.A.	94 (68.1)	N.A.	49 (35.5)	3 (6.1)	4/94 (4.3)

Legend: No.—number; Min.—minimum; Max.—maximum; SD—standard deviation, N.A.—not applicable.

**Table 2 antibiotics-10-01458-t002:** Antimicrobial resistance profile of the tested *S. aureus* strains (*n* = 49) according to their isolation sources.

Antimicrobial		No. of Tested *S*. *aureus* Strains according to Their Origin						Total
		Sheep Milk					Cow Milk	
Class	Agent	Telemea	Caș	Burduf	Urdă	Telemea	Caș	
		No. of Resistant *S. aureus*/No. of Total *S. aureus* Strains Tested (%)						
β-lactams	PCG	15/24 (62.5%)	2/6 (33.3%)	1/2 (50%)	1/1 (100%)	5/8 (62.5%)	2/8 (25%)	26/49 (53.1%)
	OXA	6/24 (25%)	1/6 (16.7%)	0/2 (0%)	0/1 (0%)	0/8 (0%)	1/8 (12.5%)	8/49 (16.3%)
	AMP	1/16 (6.3%)	0/2 (0%)	N.A.	N.A.	0/8 (0%)	0/3 (0%)	1/29 (3.4%)
	IPM	3/18 (16.7%)	1/6 (16.7%)	0/2 (0%)	0/1 (0%)	0/5 (0%)	0/6 (0%)	4/38 (10.5%)
	SAM	0/10 (0%)	0/2 (0%)	N.A.	N.A.	0/5 (0%)	0/1 (0%)	0/18 (0%)
	CET	0/6 (0%)	N.A.	N.A.	N.A.	0/3 (0%)	0/2 (0%)	0/11 (0%)
	CTF *	4/6 (66.7%)	N.A.	N.A.	N.A.	3/3 (100%)	1/2 (50%)	8/11 (72.7%)
	CEF	0/6 (0%)	N.A.	N.A.	N.A.	0/3 (0%)	0/2 (0%)	0/11 (0%)
aminoglycosides	GEN	1/24 (4.2%)	0/6 (0%)	0/2 (0%)	0/1 (0%)	0/8 (0%)	1/8 (12.5%)	2/49 (4.1%)
	KAN	7/16 (43.8%)	0/2 (0%)	N.A.	N.A.	3/8 (37.5%)	2/3 (66.7%)	12/29 (41.4%)
	AMK	5/6 (83.3%)	N.A.	N.A.	N.A.	3/3 (100%)	2/2 (100%)	10/11 (90.9%)
	NEO *	3/6 (50%)	N.A.	N.A.	N.A.	3/3 (100%)	1/2 (50%)	7/11 (63.6%)
quinolones	CIP *	2/8 (25%)	2/4 (50%)	0/2 (0%)	0/1 (0%)	N.A.	2/5 (40%)	6/20 (30.0%)
	MXF	0/8 (0%)	0/4 (0%)	0/2 (0%)	0/1 (0%)	N.A.	0/5 (0%)	0/20 (0%)
	ENR	15/16 (93.8%)	2/2 (100%)	N.A.	N.A.	5/8 (62.5%)	3/3 (100%)	25/29 (86.2%)
	MBX	0/10 (0%)	0/2 (0%)	N.A.	N.A.	0/5 (0%)	0/1 (0%)	0/18 (0%)
steroids	FA	3/18 (16.7%)	2/6 (33.3%)	0/2 (0%)	0/1 (0%)	0/5 (0%)	0/6 (0%)	5/38 (13.2%)
glycopeptides	TEC *	2/8 (25%)	1/4 (25%)	0/2 (0%)	0/1 (0%)	N.A.	2/5 (40%)	5/20 (25.0%)
	VAN	2/8 (25%)	1/6 (16.7%)	0/2 (0%)	0/1 (0%)	0/5 (0%)	0/6 (0%)	3/38 (7.9%)
lincomycins	CLI	8/24 (33.3%)	2/6 (33.3%)	0/2 (0%)	0/1 (0%)	3/8 (37.5%)	2/8 (25%)	15/49 (30.6%)
macrolides	ERY	6/24 (25%)	1/6 (16.7%)	0/2 (0%)	0/1 (0%)	3/8 (37.5%)	1/8 (12.5%)	11/49 (22.4%)
	TMS	2/6 (33.3%)	N.A.	N.A.	N.A.	1/3 (33.3%)	1/2 (50%)	4/11 (36.4%)
	TYL	1/6 (16.7%)	N.A.	N.A.	N.A.	0/3 (0%)	1/2 (50%)	2/11 (18.2%)
oxazolidinones	LZD	2/8 (25%)	1/4 (25%)	0/2 (0%)	0/1 (0%)	N.A.	0/5 (0%)	3/20 (15.0%)
phosphonic acid derivative	FOF	0/8 (0%)	1/4 (25%)	0/2 (0%)	0/1 (0%)	N.A.	0/5 (0%)	1/20 (5.0%)
pseudomonic acid derivative	MUP *	1/10 (10%)	0/2 (0%)	N.A.	N.A.	0/5 (0%)	0/1 (0%)	1/18 (5.6%)
rifampicins	RIF *	13/18 (72.2%)	2/6 (33.3%)	0/2 (0%)	0/1 (0%)	0/5 (0%)	0/6 (0%)	15/38 (39.5%)
nitrofuran derivate	NIT	0/10 (0%)	0/2 (0%)	N.A.	N.A.	0/5 (0%)	0/1 (0%)	0/18 (0%)
amphenicols	CHL	0/10 (0%)	0/2 (0%)	N.A.	N.A.	0/5 (0%)	0/1 (0%)	0/18 (0%)
	FLO	0/6 (0%)	N.A.	N.A.	N.A.	0/3 (0%)	0/2 (0%)	0/11 (0%)
sulfonamides	SXT	0/24 (0%)	0/6 (0%)	0/2 (0%)	0/1 (0%)	0/8 (0%)	0/8 (0%)	0/49 (0%)
tetracyclines	TET	11/24 (45.8%)	1/6 (16.7%)	0/2 (0%)	0/1 (0%)	6/8 (75%)	1/8 (12.5%)	19/49 (38.8%)
	TGC	0/8 (0%)	0/4 (0%)	0/2 (0%)	0/1 (0%)	N.A.	0/5 (0%)	0/20 (0%)

Legend: PCG—benzylpenicillin; OXA—oxacillin; AMP—ampicillin; IPM—imipenem; SAM—ampicillin/sulbactam; CET—cefalotin; CTF—ceftiofur; CEF—cefquinome; GEN—gentamicin; KAN—kanamycin; AMK—amikacin; NEO—neomycin; CIP—ciprofloxacin; MXF—moxifloxacin; ENR—enrofloxacin; MBX—marbofloxacin; FA—fusidic acid; TEC—teicoplanin; VAN—vancomycin; CLI—clindamycin; ERY—erythromycin; TMS—tilmicosin; TYL—tylosin; LZD—linezolid; FOF—fosfomycin; MUP—mupirocin; RIF—rifampicin; NIT—nitrofurantoin; CHL—chloramphenicol; FLO—florfenicol; SXT—trimethoprim—sulfamethoxazole; TET—tetracycline; TGC—tigecycline; N.A.—not applicable; * antimicrobials expressed intermediate MIC results, which were counted as resistant—RIF (*n* = 11), CTF (*n* = 8), NEO (*n* = 5), CIP (*n* = 4), MUP (*n* = 1), and TEC (*n* = 1);.

**Table 3 antibiotics-10-01458-t003:** Multi-drug resistance phenotype combination of the tested *S. aureus* isolates.

Cheese Assortments and Their Milk Origin		No. of Isolates	Resistance to Antimicrobial Profile of *S. Aureus* Strains (No.)	No. of Classes with Resistance	Classes with Resistance
Sheep milk	telemea	1	PCG, OXA, IPM, FA, TEC, VAN, CLI, ERY, LZD, RIF, TET (11)	8	β-lactams, glycopeptides, steroids, lincomycins, macrolides, oxazolidinones, rifampicins, tetracyclines
		1	PCG, OXA, AMP, IPM, ENR, VAN, FA, MUP, RIF, TET (10)	7	β-lactams, quinolones, glycopeptides, steroids, pseudomonic acid derivative, rifampicins, tetracyclines
		2	PCG, CTF, KAN, AMK, NEO, ENR, ERY, CLI, TET (9)	6	β-lactams, aminoglycosides, quinolones, macrolides, lincomycins, tetracyclines
		2	PCG, CTF, KAN, AMK, ENR, ERY, CLI, TET (8)		
		1	PCG, OXA, GEN, KAN, AMK, NEO, ENR, TMS, TYL, CLI (10)	5	β-lactams, aminoglycosides, quinolones, macrolides, lincomycins
		1	OXA, FA, CLI, LZD, RIF (5)		
		1	PCG, CLI, ERY, RIF, TET (5)		β-lactams, lincomycins, macrolides, rifampicins, tetracyclines
		2	PCG, ENR, TET, RIF (4)	4	β-lactams, quinolones, rifampicins, tetracyclines
		1	PCG, KAN, ENR, TET (4)		β-lactams, aminoglycosides, quinolones, tetracyclines
		1	PCG, OXA, ENR, TMS (4)	3	β-lactams, quinolones, macrolides
		1	PCG, CIP, RIF (3)		β-lactams, quinolones, rifampicins
		1	PCG, OXA, IPM, CIP, RIF (5)		
		1	PCG, KAN, TET (3)		β-lactams, aminoglycosides, tetracyclines
	caș	1	PCG, OXA, IPM, FA, TEC, VAN, CLI, ERY, LZD, RIF (10)	7	β-lactams, glycopeptides, steroids, lincomycins, macrolides, oxazolidinones, rifampicins
		1	PCG, FA, FOF (3)	3	β-lactams, steroids, phosphonic acid derivative
		1	CIP, CLI, RIF (3)		quinolones, lincomycins, rifampicins
Cow milk	telemea	1	PCG, KAN, AMK, NEO, ENR, ERY, TMS, CLI, TET (9)	6	β-lactams, aminoglycosides, quinolones, macrolides, lincomycins, tetracyclines
		2	PCG, KAN, AMK, NEO, ENR, ERY, CLI, TET (8)		
	caș	1	PCG, CTF, KAN, AMK, NEO, ENR, ERY, CLI, TET (9)	6	β-lactams, aminoglycosides, quinolones, macrolides, lincomycins, tetracyclines
		1	PCG, OXA, GEN, KAN, AMK, ENR, TMS, TYL, CLI (9)	5	β-lactams, aminoglycosides, quinolones, macrolides, lincomycins

Legend: PCG—benzylpenicillin; OXA—oxacillin; AMP—ampicillin; IPM—imipenem; SAM—ampicillin/sulbactam; GEN—gentamicin; KAN—kanamycin; AMK—amikacin; NEO—neomycin; ENR—enrofloxacin; FA—fusidic acid; TEC—teicoplanin; VAN—vancomycin; CLI—clindamycin; ERY—erythromycin; TMS—tilmicosin; TYL—tylosin; LZD–linezolid; FOF—fosfomycin; MUP—mupirocin; RIF—rifampicin; FLO—florfenicol; TET—tetracycline.

## Data Availability

All data generated or analyzed during this study are included in the submitted version of the manuscript.

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
