# Peer review of "Multidrug Resistant Coagulase-Positive *Staphylococcus aureus* and Their Enterotoxins Detection in Traditional Cheeses Marketed in Banat Region, Romania"

_antibiotics, 2021, doi:10.3390/antibiotics10121458_

Round 1
Reviewer 1 Report
The authors investigated the presence and public health significance of the CPS in traditional cheese products. The study is well designed, material and methods and results clearly exposed, but I suggest to:
- Indicate the entire name of bacteria and then the acronym both in the Abstract and Introduction sections,
- Add the aim of the study in the Introduction since it is missing.
Author Response
Reviewer #1:
The authors investigated the presence and public health significance of the CPS in traditional cheese products. The study is well designed, material and methods and results clearly exposed, but I suggest to:
Answer: Thank you very much for your overall positive appreciations!
Comment (1) Indicate the entire name of bacteria and then the acronym both in the Abstract and Introduction sections
Answer (1) The requested changes has been operated in the abstract section.
Comment (2) Add the aim of the study in the Introduction since it is missing.
Answer (2) The study aim is retrieved at the end of the introduction section, highlighted with red font between lines 80-85
Reviewer 2 Report
This manuscript focuses on the identification of coagulase-positive and multidrug-resistant Staphylococcus aureus from traditional cheeses marketed in Banat region, Romania. Generally, it is well written and has certain practical applications for the biological safety of cheese products. However, the data presented are preliminary and some major revisions are suggested. Detailed comments are as below.
(1) Line 36: Data for methicillin-resistant Staphylococcus aureus were not presented in the Results part.
(2) Lines 38-39: This sentence should be rewritten since it is similar to that in lines 356-358.
(3) Line 84: Why not detect other antibiotic resistance genes in Staphylococcus aureus?
(4) Line 86: The Results part should be divided into different sections (e.g., CPS detection, molecular characteristics, enterotoxin detection) in accordance with the Materials and Methods part.
(5) Lines 100-102: What are specific characteristics (e.g., genomic features) of these three isolates harboring the mecA gene?
(6) Lines 105-115: Why did you use different cards (e.g., AST-GP79, AST-P592 and AST-GP69) for antibiotic susceptibility determination? There seem to be some discrepancies in antibiotic resistance phenotype assessed by different cards.
(7) Lines 164-166: Please add some references.
(8) Lines 230-232: More in-depth discussions are required.
(9) Lines 361-362: Please elaborate on future research perspectives.
Author Response
Dr. Karina Yang,
Section Managing Editor
Antibiotics-Basel
Timisoara, 15/11/2021,
Romania
Dear Editor,
We would like to thank the Editor and each reviewer for their time, interest and comments to our manuscript. We are delighted to be considered for publication in the prestigious Antibiotics-Basel journal and to receive quality peer reviews that have greatly improved our manuscript. We have thoroughly considered each Reviewer's comments and have revised the text to reflect this. We have addressed each Reviewer’s comments below, in a point-by-point fashion. The new version of the manuscript contains all of the requested and operated changes and additional information (as reviewers required) highlighted with red font.
Below, you can find the answers to the all comments and suggestions.
Thank you again for your time and consideration.
Sincerely yours,
Dr. Kálmán IMRE
On behalf of all the authors
Reviewers' comments:
Reviewer #2:
This manuscript focuses on the identification of coagulase-positive and multidrug-resistant Staphylococcus aureus from traditional cheeses marketed in Banat region, Romania. Generally, it is well written and has certain practical applications for the biological safety of cheese products. However, the data presented are preliminary and some major revisions are suggested. Detailed comments are as below.
Answer: Thank you very much for your overall positive appreciations!
Comment (1) Line 36: Data for methicillin-resistant Staphylococcus aureus were not presented in the Results part.
Answer (1) From the original submitted version the sentence “From the total number of isolates, three (6.1%; 95% CI 2.1-16.5) harbored the mecA gene, being recovered from two (8.3%) telemea and one (16.6%) caș assortments (Table 1).” referred to the occurrence of the methicillin-resistant S. aureus strains. However, to be clearer, this section was completed with the following sentence: “These isolates were accounted as methicillin-resistant, because the detection of this gene represents the standard procedure in the determining resistance towards methicillin.”
Comment (2) Lines 38-39: This sentence should be rewritten since it is similar to that in lines 356-358.
Answer (2) According to the reviewer requirement the sentence was rewritten resulting in: „The investigation highlighted a common occurrence of multidrug-resistant S. aureus strains in the monitored cheese assortments, which can constitute a potential risk for consumers’ health.”
Comment (3) Line 84: Why not detect other antibiotic resistance genes in Staphylococcus aureus?
Answer (3): During investigations the authors were aware that beside phenotypic resistance screening of the isolated strains, their findings need to complete with some molecular tools in order to publishable in high impact factor journals. Considering the existent limited financial resources, they have chosen to molecularly confirm of the isolated S. aureus strains using a genus and species-specific primer set, and out from the antibiotic resistance genes the mecA gene, because the detection of this gene represents the standard procedure in the determining resistance towards methicillin. MRSA is resistant, not only to methicillin (methicillin and oxacillin), but also to other categories of antimicrobials as macrolides, chloramphenicol, aminoglycosides, tetracyclines, and lincosamides, so it causes a threat to public health because it is difficult to treat making patients’ treatment options limited and made the search for new compounds active against it inevitable (https://doi. org/10.1155/2019/8321834). However, this limitation of the study was highlighted in the conclusion section as future perspective.
Comment (4) Line 86: The Results part should be divided into different sections (e.g., CPS detection, molecular characteristics, enterotoxin detection) in accordance with the Materials andMethods part.
Answer (4): According to the reviewer requirement the Results section was divided in subheading in accordance with the Materials and Methods section resulting in:
2.1. Frequency of isolation and contamination level of CPS, and incidence of S. aureus
2.2. Prevalence of genetic markers among the isolated S. aureus strains
2.3. Detection of SEs in cheese samples
2.4. Antimicrobial resistance profile of S. aureus isolated strains
Comment (5) Lines 100-102: What are specific characteristics (e.g., genomic features) of these three isolates harbouring the mecA gene?
Answer (5): According to the reviewer requirement the following sentence was inserted: “It is important to mention that in case of this MRSA strains, out from the total genome, ~25% can consist on the presence of mobile genetic elements (e. g. bacteriophages, plasmids, transposoms, pathogenecity islands or chromosomal cassettes) constituting the so-called accessory genome, responsible for genetic material transfer between strains.”
Comment (6) Lines 105-115: Why did you use different cards (e.g., AST-GP79, AST-P592 and AST-GP69) for antibiotic susceptibility determination? There seem to be some discrepancies in antibiotic resistance phenotype assessed by different cards.
Answer (6): As has been highlighted within the line 254 of the Materials and methods section, the study was conducted within two distinct period, meaning February 2012 and October 2013, and January 2020 to February 2021. Accordingly, as has been mentioned within the lines 321-323 of the same section, the specific Gram-positiv bacteria cards were selected in the study periods depending by their commercial availability by the Romanian distributor and aiming to cover both livestock and human origin antimicrobials. In this regards, within the period February 2012 – October 2013 we used the kits AST-P592 including human drugs and the kit AST-GP69 including veterinary drugs. Furthermore, in the period January 2020 – February 2021 the card AST-GP79 including veterinary drugs. We included this information’s in the Titles of the tables 2-4 incorporated as supplementary files.
Comment (7) Lines 164-166: Please add some references.
Answer (7): As requested the references 4, 16 and 17 were inserted.
Comment (8) Lines 230-232: More in-depth discussions are required.
Answer (8): As the reviewer requested, the following sentences were inserted to improve the discussion of the recorded MDR of the tested S. aureus strains.
The recorded high number of MDR S. aureus isolates (49%) is another notable finding of the present survey. In this regards, it is important to mention that the consideration of intermediate MIC results as resistant towards a certain drug resulted in the reporting of more resistance within the tested S. aureus strains. Our results, in general, confirm the steady upward trend of AMR phenomenon reported in other earlier studies for food and/or animal origin strains. Thus, in the neighboring Sebia, 25.0% of the tested isolates expressed MDR and the most frequently encountered resistance profiles included PCG, CIP or GEN [4]. In another investigation, a total of 12.8% of the screened milk and dairy products origin S. aureus isolates were MDR [26]. In addition, in context of the One Health approach, the recorded worrying 100% MDR pattern in case of S. aureus strains isolated from shelter dogs with skin lesions from Timișoara Municipility, included in the monitored Banat region in the present study, indirectly suggest sthe increased AMR trend in case of this pathogen in Romania [36]. In contrats, in a study conducted in Poland most of the isolates (47.5%) has been found resistant to only one antimicrobial [16].
Comment (9) Lines 361-362: Please elaborate on future research perspectives.
Answer (9): According to the reviewer suggestion the authors inserted the following sentence representing as future perspective: “The present study offer baseline information on the knowledge of the phenotypic antimicrobial resistance of food origin S. aureus strains in the screened region. However, further studies based on molecular tools, investigating the presence of other antibiotic resistance genes and mobile genetic elements responsible for genetic material transfer between S. aureus strains, are still necessary to a better knowledge of the complex puzzle AMR phenomenon of this important food-borne pathogen in Romania.”
Reviewer 3 Report
The study goal was to determine the prevalence of coagulase positive staphylococci and antimicrobial susceptibility profile of S. aureus isolates in local cheese produced from raw milk. The study addresses a food safety concern. While there are several previous studies were conducted on this topic in many countries, this might be of interest to the local health authorities. Below are my technical comments on this manuscript:
Materials and Methods
1. Please justify your choice of number of samples.
2. Why using old samples (2012 - 2013) and then new from 2020? How the old samples were handled? There is no results or discussion about the sampling year. Major issue!
3. How the nine agri-food markets were selected? Sampling plan?
4. How the samples were collected "randomly"? It is not enough to just say "randomly" without explaining the randomization method.
5. The authors considered intermediate MIC results as "resistant". It should be considered susceptible and not resistant. Another major issue that resulted in reporting more resistance that it should be.
6. Data analyses: What about the count cfu data? how were they analyzed?
Results
- There is no statistical analyses for the AMR data to compare between sample type.
- The tables 2 - 4 should be changed into two tables that shows the resistance prevalence by sample type and antibiotics and another table that shows frequency of the multi-drug resistant phenotype combination.
- Why there was no determination of MRSA?
Author Response
Dr. Karina Yang,
Section Managing Editor
Antibiotics-Basel
Timisoara, 15/11/2021,
Romania
Dear Editor,
We would like to thank the Editor and each reviewer for their time, interest and comments to our manuscript. We are delighted to be considered for publication in the prestigious Antibiotics-Basel journal and to receive quality peer reviews that have greatly improved our manuscript. We have thoroughly considered each Reviewer's comments and have revised the text to reflect this. We have addressed each Reviewer’s comments below, in a point-by-point fashion. The new version of the manuscript contains all of the requested and operated changes and additional information (as reviewers required) highlighted with red font.
Below, you can find the answers to the all comments and suggestions.
Thank you again for your time and consideration.
Sincerely yours,
Dr. Kálmán IMRE
On behalf of all the authors
Reviewer #3:
The study goal was to determine the prevalence of coagulase positive staphylococci and antimicrobial susceptibility profile of S. aureus isolates in local cheese produced from raw milk. The study addresses a food safety concern. While there are several previous studies were conducted on this topic in many countries, this might be of interest to the local health authorities.
Answer: Thank you very much for your overall positive appreciations!
Below are my technical comments on this manuscript:
Comment (1). Please justify your choice of number of samples.
Answer (1): During investigations the research team tried to monitor the microbiological quality regarding the presence of CPS, including S. aureus, in traditional cheeses of as many of possible local cheese producers. The number of totally processed products from each assortment can be considered approximately directly proportional with the amount of the commercialized products, according to the consumer preferences in the Banat area. Thus, the production and consumption of telemea cheese is considered three times larger than caș assortment, which in turn is three times more available than urdă and burduf cheeses. As has been highlighted in the original submitted version of the manuscript, a total of 9 agri-food markets were selected. Samples were harvested from a total of 42 producers, during a total of 10 visits, within the 10 trimesters of the study period.
- 109 telemea cheese samples were provided from = 10 cheese producers, each of them was sampled during the 10 visits (meaning 100 samples) + another 9 cheese producers, each of them being sampled within a single visit (meaning other 9 samples)
- 38 caș samples were provided from = 3 cheese producers, each of them was sampled during the 10 visits (meaning 30 samples) + another 8 cheese producers, each of them being sampled within a single visit (meaning other 9 samples);
- 11 burduf cheese samples were provided from = 1 cheese producer sampled during the 10 visits (meaning 10 samples) + another producer harvested only once;
- 11 urdă samples came from 11 different producers
Comment (2). Why using old samples (2012 - 2013) and then new from2020? How the old samples were handled? There is no results or discussion about the sampling year. Major issue!
Answer (2): The manuscript presents the investigation results conducted in two different periods, February 2012 and October 2013, and January 2020 and February 2021. In the mentioned two periods all investigations (isolation and enumeration of coagulase-positive staphylococci (CPS) and identification of S. aureus, molecular analyses, enterotoxin detection, antimicrobial susceptibility testing) have been completely done, and no any strain isolated in the period February 2012 - October 2013 were processed/reanalyzed in the priod 2020 and 2021.
According to the rewiever requirement, relevant informations about the obtained results according to the two sampling periods and its corresponding discussions were included.
“Of the 169 traditional raw milk origin cheese samples examined, 138 (81.6%; 95% CI 75.1 – 86.7) were contaminated with CPS, with a distribution of 101 (83.5% from the total of 121; 95% CI 75.8 – 89.0) and 37 (77.1% from the total of 48; 95% CI 63.5 – 86.7) positive samples in the periods of February 2012 – October 2013, and January 2020 – February 2021, respectively.”
“Out of them, 38 (37.6% from the total of 101; 95% CI 28.8 – 47.4) samples were obtained in the study period February 2012 – October 2013, and another 11 (29.7% from total of 37; 95% CI 17.5 – 45.8) samples in the period January 2020 – February 2021, with an isolation frequency of 37.6% and 29.7%, respectively”
“From the total number of isolates, three (6.1%; 95% CI 2.1-16.5) harbored the mecA gene, being recovered from two (8.3%) telemea and one (16.6%) caș assortments in the period of January 2020 – February 2021 (Table 1)”
“All of these positive findings were obtained in the study period of February 2012 – October 2013.”
“In addition, it is important to mention that the lack of the statistically significant associations (p<0.05) between the isolation frequencies of CPS and S. aureus in the two study periods (February 2012 – October 2013 vs. January 2020 – February 2021), can reflect the lack of the hygienic measures improvements during the last ten years at the level of small scale integrated dairy production units, even if important financial resources were attrachted by farmers within the European Agricultural Fund for Rural Development program.”
Comment (3) How the nine agri-food markets were selected? Sampling plan?
Answer (3): To clarify this issue the following sentence was inserted: “The selected agri-food markets were located in the two county (Timiș and Caraș-Severin) residences of the screened region, namely Timișoara (n=6) and Reșița (n=3) Municipalities, being visited daily by hundred of consumers. The samples were aseptically collected by the research team members, in the first month of each trimester of the calendar year, accompanying the veterinary authorities during the officially organized control visits.”
Comment (4) How the samples were collected "randomly"? It is not enough to just say "randomly" without explaining the randomization method.
Answer (4): The following sentence was inserted in order to explain the randomization method: “Sample collection followed the requirement of the European Regulation (EC) No. 2073/2005, meaning that for each sample, a pool of the randomly selected five products, from the same category and retailer were collected.”
Comment (5). The authors considered intermediate MIC results as "resistant". It should be considered susceptible and not resistant. Another major issue that resulted in reporting more resistance that it should be.
Answer (5): With respect to the reviewer opinion the authors would like to maintain their consideration to account intermediate MIC results as resistant, mainly from the following two consideration:
- the susceptibility of a bacterial strain to a given antibiotic is considered to be intermediate when it is inhibited in vitro by a concentration of this drug that is associated with an uncertain therapeutic effect;
- the intermediate category most frequently signals need for a dose higher than the standard dose, favouring the development of antibiotic resistance.
Thus, from the clinical point of view, the categorization of bacterial strains with an intermediate resistance level as resistant, resulting in their exclusion in the management of infections, will lead to the avoiding of uncertain therapeutic effect and possible acquiring of antimicrobial resistance. Several recently published articles in Antibiotics (Basel) presenting the results of the monitoring of antimicrobial susceptibility profile of S. aureus isolates adopted this strategy (e.g. PMID:33027900; PMID: 34680784; PMID: 33106494).
However, in the Discussion section the authors referred to the concern raised by the reviewer, mentioning that the consideration of intermediate MIC results as “resistant” resulted in the reporting of more resistance of the tested strains (see the lines 257-259 of the revised version).
Comment (6) Data analyses: What about the count cfu data? how were they analysed?
Answer 6: The number of typical colonies was accounted using the manual counting of colony-forming (CFU) units’ technique, recommended to estimate the number of viable bacteria cells. The obtained results of the frequency of isolation and contamination level of CPS, as well as discussions related by these finding were presented within the lines 89-99, and 178-200, respectively.
Results
Comment (7) There is no statistical analyses for the AMR data to compare between sample type.
Answer (7): In the original submitted version of the manuscript the authors stated that “No notable associations (p<0.05) were registered between the expressions of the antimicrobial resistance patterns of the tested S. aureus isolates and their isolation source.”
Comment (8) The tables 2 – 4 should be changed into two tables that shows the resistance prevalence by sample type and antibiotics and another table that shows frequency of the multi-drug resistant phenotype combination.
Answer (8): Special thanks for this suggestion!
According to the reviewer requirement two new tables were elaborated in the revised version of the manuscript. However, in order to mention the overview of the two other reviewers about the data presentation, and to inform the reader about the included antimicrobials in each card, and the used MIC range values in case of each antimicrobial, the results presented in Tables 2-4 were included in a separate document as supplementary files.
Comment (9) Why there was no determination of MRSA?
Answer (9): In the results section the following sentences presents the methicillin-resistant S. aureus results: „From the total number of isolates, three (6.1%; 95% CI 2.1-16.5) harbored the mecA gene, being recovered from two (8.3%) telemea and one (16.6%) caș assortments (Table 1). These isolates were accounted as methicillin-resistant, because the detection of this gene represents the standard procedure in the determining resistance towards methicillin.”
Round 2
Reviewer 2 Report
The revised manuscript is acceptable in its current form.
Author Response
Thank you for you suggestions, time and overall positive appreciations!
Reviewer 3 Report
The authors failed to address properly the major issues I raised with the study. In addition, there are serious flaws in the study design. Specifically:
- The authors did not provide the sample size calculation. It is not enough to say how the samples were collected.
- The number of samples (10 years apart) is very small to make comparison between the safety of the product before and after 10 year period
- The authors failed to explain the randomization process in their sampling plan.
- Inflating the phenotypic resistance percentages due to including intermediate MIC as resistant is a problem for the cheese producers and might have serious implications.
Author Response
The authors failed to address properly the major issues I raised with the study. In addition, there are serious flaws in the study design. Specifically:
- The authors did not provide the sample size calculation. It is not enough to say how the samples were collected.
According to the reviewer requirement, the authors provided details about sample size calculation. The following sentences were inserted:
“The number of cheese producers to be sampled was determined based on the evidence of their total number by Sanitary Veterinary Authority within the study periods in the screened area, using the online version of GeoPoll Sample Size Calculator (https://www.geopoll.com/blog/sample-size-research/) with 95% confidence level, and 5% absolute precision (confidence interval). The required number of cheese producers was determined to be 42. In addition, the total number of the collected samples (n=169) resulted from the availability and diversity of the sold cheese assortments by producers during sampling visits and taking into consideration the fulfillment of the Romanian National Order No.13/2005 requirement [36].”
- The number of samples (10 years apart) is very small to make comparison between the safety of the product before and after 10 years period.
During the first revision the reviewer comment was “There is no results or discussion about the sampling year.” According to this requirement, relevant information about the obtained results according to the two sampling periods and its corresponding discussions were included. Concerning the raised concern by the reviewer, the inserted sentences in the discussion chapter namely „In addition, it is important to mention that the lack of the statistically significant associations (p<0.05) between the isolation frequencies of CPS and S. aureus in the two study periods (February 2012 – October 2013 vs. January 2020 – February 2021), can reflect the lack of the hygienic measures improvements during the last ten years at the level of small scale integrated dairy production units, even if important financial resources were attracted by farmers within the European Agricultural Fund for Rural Development program.”, in the authors opinion are scientifically corrects, presenting statistically valid information and reflects the reality.
- The authors failed to explain the randomization process in their sampling plan.
During the first revision, the authors inserted the following sentence referring to the randomized sampling plan: “Sample collection followed the requirement of the European Regulation (EC) No. 2073/2005, meaning that for each sample, a pool of the randomly selected five products (~300 g), from the same category and retailer were collected.”
For instance, in a case of a cheese producer selling telemea cheese, there are several cheese pieces (units) exposed to selling within the agri-food market. During sampling the authors randomly (without a conscious decision/indiscriminately/ in a way that involves equal chances for each item.) selected five pieces/units (~300 g) resulting a pool of five units comprising the one sample, from the same assortments.
To be more concluding, this information (the sampling methodology) is included on documents retrieved on the following links:
Pg. 20/30 – M1 - 2.2.3 in
https://eur-lex.europa.eu/legal-content/EN/TXT/PDF/?uri=CELEX:02005R2073-20140601&from=DA
and
Pg. 22/27– Art. 30, Letter H, 1-3 paragraph.
The sampling procedure exactly follow the regulations in force.
- Inflating the phenotypic resistance percentages due to including intermediate MIC as resistant is a problem for the cheese producers and might have serious implications.
With respect to the reviewer opinion, within the first review round the authors provided detailed and scientifically sustained answers, motivating their choice in considering intermediate MIC results as resistant. In addition, the authors would like to underline the fact that, as has been mentioned in the materials and methods section, the sampling was conducted in presence of veterinary authorities during the officially organized control visits, resulting in the communication of final results to them. So, they are aware that the results of the present study reflect the worrying current situation, actually valid for the whole country. Furthermore, the publishing of the results of current survey is the authors solely responsibility. Last but not least, the authors want to highlight other four considerations:
- the transforming intermediate resistant to susceptible will result in a completely new manuscript data, altering the positive overview of other two reviewers, giving their final approval for the acceptance of the manuscript;
- considering intermediate resistance level as susceptible, will mislead the physicians and veterinarians in the management of S. aureus infections;
- in the research team opinion, for a successful treatment of an infection the practician never choose antimicrobial with an intermediate resistance level;
- searching through the scientific literature, no published scientific paper has been recorded in which intermediate resistance level results were interpreted (considered) as susceptible;
However, we will defer to the editor and will transform the “intermediate” in “susceptible” if it is determined that it is needed and appropriate.
Thank you for your understanding!
Round 3
Reviewer 3 Report
The study still have major issues that need to be addressed.
- The sample size was not calculated properly. Sample size calculation for a single proportion require a prior estimate of a "prevalence" to use in the formula. Trying to calculate the sample in retrospective is not the way to go!
- Again, using a small sample size to look at two study periods (February 2012 – October 2013 vs. January 2020 – February 2021) that are 10 year apart does NOT give you enough power to compare. What is the point of this compare contamination prevalence of few samples that 10 year apart!
- Since the authors disagree with including the intermediate MICs with susceptible MICs, then the only solution is to present the findings in three different categories (susceptible, intermediate, and resistant).
Author Response
Dear Reviewer,
The research team answers to the remained concerns are presented below:
Comment 1: The sample size was not calculated properly. Sample size calculation for a single proportion require a prior estimate of a "prevalence" to use in the formula. Trying to calculate the sample in retrospective is not the way to go!
Answer 1: In the lack of any previously conducted study in Romania aiming to determine the occurrence of coagulase positive staphylococci in traditional cheeses, the expected prevalence was considered the one reported in the neighboring Serbia, meaning 87.4%. Thus, the required total number of samples to be investigated was computed using the formula proposed by Cochran W.G. (1963) as follow:
P=87,4% / 0,874
5% / 0,05
95% / 1,96
n=169
|
Cochran |
|||
|
p |
0,874 |
n=169,2209 |
|
|
t |
1,96 |
||
|
D |
0,05 |
The proposed formula by Cochran, W.G. was:
n – sample size;
t – significance of the test coefficient;
p – expected prevalence;
∆ - maximum admitted error.
Reference: Cochran, W.G. Sampling Technique. 1963, 2nd Edition, John Wiley and Sons Inc., New York.
In conclusion, in the manuscript the following sentence was inserted:
„The total number of samples to be collected (n=169) was determined using the proposed formula by Cochran (1963), with 87.4% an expected prevalence, 5% absolute precision (confidence level), and 95% confidence interval (CI) [36].”
Comment 2: Again, using a small sample size to look at two study periods (February 2012 – October 2013 vs. January 2020 – February 2021) that are 10 year apart does NOT give you enough power to compare. What is the point of this compare contamination prevalence of few samples that 10 year apart!
Answer 2: In order to clarify this issue, the authors want to highlight the fact that the original submitted version of he manuscript did not contain any comparation between the study periods. However, during the first revision round the reviewer request was: „There is no results or discussion about the sampling year. Major issue!” To fulfill this concern, we have inserted the presentation of the most important results in the result section, as well as a short discussion paragraph as follow:
Results
“Of the 169 traditional raw milk origin cheese samples examined, 138 (81.6%; 95% CI 75.1 – 86.7) were contaminated with CPS, with a distribution of 101 (83.5% from the total of 121; 95% CI 75.8 – 89.0) and 37 (77.1% from the total of 48; 95% CI 63.5 – 86.7) positive samples in the periods of February 2012 – October 2013, and January 2020 – February 2021, respectively.”
“Out of them, 38 (37.6% from the total of 101; 95% CI 28.8 – 47.4) samples were obtained in the study period February 2012 – October 2013, and another 11 (29.7% from total of 37; 95% CI 17.5 – 45.8) samples in the period January 2020 – February 2021, with an isolation frequency of 37.6% and 29.7%, respectively”
“From the total number of isolates, three (6.1%; 95% CI 2.1-16.5) harbored the mecA gene, being recovered from two (8.3%) telemea and one (16.6%) caș assortments in the period of January 2020 – February 2021 (Table 1)”
“All of these positive findings were obtained in the study period of February 2012 – October 2013.”
Discussion
“In addition, it is important to mention that the lack of the statistically significant associations (p<0.05) between the isolation frequencies of CPS and S. aureus in the two study periods (February 2012 – October 2013 vs. January 2020 – February 2021), can reflect the lack of the hygienic measures improvements during the last ten years at the level of small scale integrated dairy production units, even if important financial resources were attracted by farmers within the European Agricultural Fund for Rural Development program.”
Even if in the authors opinion the aforementioned comparation is scientifically correct and statistically valid, to avoid any confusion and subsequent different opinions with the reviewer, the authors chose to withdraw this complex sentence from the discussion chapter.
Comment 3: Since the authors disagree with including the intermediate MICs with susceptible MICs, then the only solution is to present the findings in three different categories (susceptible, intermediate, and resistant).
Answer 3: Taking into account the Academic Editor recommendation, namely:
“One of the reviewers raises some important points that should be better considered. The point raised about intermediate MICs is in my opinion important. Would it be possible to add a note in the legend of the table to indicate that the strains with intermediate MICs were counted with those resistant? Subsequently, in the discussion, would it be possible to write a better justification? You could use some of the elements already written to respond to the reviewer.”
the research team operated the following changes:
In the legend of the Table 1 the authors inserted:
„* - antimicrobials expressed intermediate MIC results, which were counted as resistant – RIF (n=11), CTF (n=8), NEO (n=5), CIP (n=4), MUP (n=1), and TEC (n=1);”
In the discussion chapter the authors inserted a justification for considering intermediate MIC resistant results as resistant „Usually, the using of a drug to which a bacterium has expressed an intermediate level of resistance in the management of infections caused by that bacterium, is associated with an uncertain therapeutic effect. Also, the intermediate category signals the need for a higher dose than the standard one, favoring the development of antibiotic resistance [1]. Based on these considerations and to avoid undesired outcomes of the treatment of S. aureus infections by physicians and veterinary practitioners, in the present study the obtained intermediate resistance levels in case of some drugs (RIF, CTF, NEO, CIP, MUP, TEC) were considered as resistant (Table 1).”
Thank you very!